# GALSTM-FDP: A Time-Series Modeling Approach Using Hybrid GA and LSTM for Financial Distress Prediction

Amal Al Ali [1,†] , Ahmed M. Khedr [2,3,*,†] , Magdi El Bannany [4,5,†] and Sakeena Kanakkayil [2,†]

1   Information Systems Department, University of Sharjah, Sharjah 27272, United Arab Emirates
2   Computer Science Department, University of Sharjah, Sharjah 27272, United Arab Emirates
3   Mathematics Department, Zagazig University, Zagazig 44519, Egypt
4   College of Business Administration, Umm Al Quwain University,
    Umm Al Quwain 536, United Arab Emirates
5   Department of Accounting and Auditing, Faculty of Business, Ain Shams University, Cairo 11566, Egypt
*   Correspondence: akhedr@sharjah.ac.ae
†   These authors contributed equally to this work.

**Abstract:** Despite the obvious benefits and growing popularity of Machine Learning (ML) technology, there are still concerns regarding its ability to provide Financial Distress Prediction (FDP). An accurate FDP model is required to avoid financial risk at the lowest possible cost. However, in the Internet era, financial data are exploding, and they are being coupled with other kinds of risk data, making an FDP model challenging to operate. As a result, researchers presented several novel FDP models based on ML and Deep Learning. Time series data is are important to reflect the multi-source and heterogeneous aspects of financial data. This paper gives insight into building a time-series model and forecasting distress far in advance of its occurrence. To build an efficient FDP model, we provide a hybrid model (GALSTM-FDP) that incorporates LSTM and GA. Unlike other previous studies, which established models that predicted distress probability only within one year, our approach predicts distress two years ahead. This research integrates GA with LSTM to find the optimum hyperparameter configuration for LSTM. Using GA, we focus on optimizing architectural aspects for modeling the optimal network based on prediction accuracy. The results showed that our algorithm outperforms other state-of-the-art methods in terms of predictive accuracy.

**Keywords:** financial distress prediction (FDP); long short term memory (LSTM); genetic algorithm (GA); machine learning (ML)

## 1. Introduction

Financial distress prediction (FDP), the significant factor of enterprise risk management, is also the core of enterprise financial distress theory (Wanke et al. 2015). As a result of the COVID-19 pandemic and current global economic recession, the probabilities of numerous sorts of businesses entering financial distress or insolvency is steadily increasing (El-Bannany et al. 2021; Khan et al. 2020). Many studies have shown that ignoring enterprise financial risks is an important cause of business failure, and FDP has a great impact on corporate sustainability (Mehreen et al. 2020). Thus, for enterprises with significant financial risks, but which have not yet caused significant losses, it is critical to be able to timely spot potential financial difficulties and warn them (Hu and Sathye 2015; Khedr et al. 2021). For businesses and other market participants, obtaining an FDP model with significant operability, strong predictive accuracy, and a broad application scope is crucial. Machine Learning (ML) approaches began to emerge as efforts in the field of FDP increased, including Neural Network (NN), Genetic Algorithm (GA), Support Vector Machines (SVM), etc. Moreover, as ML technology improves, combining several ML approaches during the FDP process has become a new trend (Sreedharan et al. 2020a; Zhu et al. 2022). The majority of studies indicate that data mining techniques can forecast distress better and outperforms

traditional methods (Sreedharan et al. 2020b). For model construction, on one hand, classic statistical and ML methods are applied in feature engineering and classification, such as Naïve Bayesian, SVM, and ensemble learning, including decision-trees-based Gradient Boosting Decision Tree (GBDT), Random Forest (RF), eXtreme Gradient Boosting (XGB), and Adaptive Boosting (AdaBoost) (Başoğlu Kabran and Ünlü 2021; Gepp and Kumar 2015; Khedr et al. 2021; Kim et al. 2019; Min and Lee 2005; Wu et al. 2008). On the other hand, various Deep Learning (DL) models are also employed for modeling, such as Genetic Algorithm (GA), Convolutional Neural Network, and Self Organizing Map (SOM) (Bukhari et al. 2020; Jang et al. 2019). Because of their excellent prediction accuracy, Neural Networks are frequently employed in FDP (Zhu et al. 2022).

Past studies on FDP show that DL techniques performed better than traditional techniques over time-series data (El Bannany et al. 2021; Geng et al. 2015; Ribeiro and Lopes 2011). In particular, among DL architectures, Recurrent Neural Networks (RNNs) have proved their superiority in analyzing financial time-series data (Bukhari et al. 2020). A recurrent neural network (RNN) is an ANN in which node connections form a directed graph along a time sequence. As a result, it can display temporal dynamic behavior. The selection of hyperparameters has a significant impact on the efficiency of an LSTM. The fine-tuning of these parameters is a critical step in increasing the model's prediction accuracy. Using brute force trial and error to try every possible combination of reasonable parameters is one method for determining ideal hyperparameters. Many LSTMs must be trained one by one, multiple times. This takes a significant amount of time and computer resources.

This paper gives insight into building a time-series model and forecasting distress far in advance of its occurrence. In order to make up for the shortcomings of a single prediction model, a new hybrid model that incorporates LSTM and GA is presented in this work. We created an FDP model using a DL algorithm called the Long Short-Term Memory (LSTM) Recurrent Neural Network (RNN). An LSTM network is a type of deep RNN model composed of LSTM units, which was introduced as a way to overcome the long-term dependency problem (Hochreiter and Schmidhuber 1997). The LSTM can scale to much longer sequences than simple RNN, overcoming the intrinsic drawbacks of simple RNN. The LSTM is advantageous for forecasting financial distress because it can efficiently learn sequential patterns in the given data including sequential or temporal characteristics and predicts time-series data well. GA is a metaheuristic and stochastic optimization algorithm inspired by the process of natural evolution, and it is widely used to find near-optimal solutions to optimization problems with large search spaces (Sun and Hui 2006). To build an efficient FDP model, we provide a hybrid model that incorporates LSTM and GA. In contrast to other previous studies that predicted financial distress probability only within one year, our GALSTM-FDP approach predicts distress two years ahead. This research integrates GA with LSTM to find the optimum hyperparameter configuration for LSTM. Using GA, we focus on optimizing architectural aspects for modeling the optimal network based on prediction accuracy. GA is a heuristic search and optimization technique based on natural selection, and it is commonly employed to obtain a near-optimal solution to optimization problems with a broad parameter space (Sun and Hui 2006). To the best of the authors' knowledge, this is the first study that uses a hybrid optimized GA and LSTM model in FDP studies based on time-series data and predicts distress more than a year in advance. Because we incorporated just within-company data, our model is compatible with any region data.

The major contributions of this paper are: First, we propose an LSTM-RNN model as an FDP model that accurately predicts distress two years ahead. The LSTM is one of the most advanced DL architectures for capturing long-term dependencies from financial time-series data. In order to improve the prediction performance of the FDP model, we provide a hybrid optimized GALSTM-FDP model. We integrate GA with LSTM to find the optimum hyperparameter configuration for LSTM where we focus on optimizing architectural aspects of the model for enhanced prediction power. Second, we provide a

strong and effective distress prediction model that is beneficial for the corporate, market participants and policymakers.

The rest of the study is structured as follows: Section 2 provides a brief overview of the related research. The proposed GALSTM-FDP model formation is presented in Section 3. Data and modeling details are provided in Section 4. Section 5 discusses the results and compares the prediction performance of the proposed hybrid model with existing ML models. Finally, Section 6 concludes the paper by stating potential future directions.

## 2. Related Research

Statistical and ML approaches have been utilized in earlier research to forecast financial distress (Sreedharan et al. 2020a). Conventional statistical models employed in the domain of distress prediction include discriminant analysis and the logit model. Altman (1968) utilized discriminant analysis to anticipate financial distress in firms, whereas Ohlson (1980) used the logit model to forecast financial distress. Later, Falbo (1991) presented a modified discriminatory model of study over a period of several years using financial stability ratios to enhance discriminatory strength. Traditional linear techniques were ineffective and had their own limits. Such straightforward approaches need more predictive history data and could not be utilized to develop a strong classifier for actual predictions. SVM as a non-parametric binary classification technique is widely used in financial time-series forecasting, and the work in (Başoğlu Kabran and Ünlü 2021) utilized SVM for the prediction of bubbles.

As financial data are considered non-linear, statistical approaches cannot be utilized to build a credible prediction model. ML methods are widely employed in financial distress prediction because of their significant benefits in extracting non-linear data relationships without previous input knowledge. As a result, ML techniques such as Logistic Regression, SVM (Min and Lee 2005; Sun et al. 2017), and NNs (Cleofas-Sánchez et al. 2016; Ravisankar and Ravi 2010) were incorporated in past research. According to their results, these approaches have proven relative effectiveness over statistical methods due to their capability to identify noisy data without making any statistical predictions. Cleofas-Sanchez et al. applied Santiago-Montero's hybrid associative classifier with translation (HACT) model to predict financial distress and provided empirical results supporting that HACT dominated four traditional neural networks, including multi-layer perceptron (MLP), radial basis function (RBF), Bayesian network (BN), and voted perceptron (VP), one SVM, and one multi-variate logistic regression (LR) model (Cleofas-Sánchez et al. 2016). While the HACT model can be trained easily due to its feed-forward learning framework, it is only suitable for simple data with repetitive structures. It would not be able to generate good learning and prediction results for complicated data. Chou et al. proposed a GA-based fuzzy clustering algorithm for FDP (Chou et al. 2017). In particular, key financial ratios selected by the GA are clustered by the fuzzy C-means clustering after the training data are divided into financially distressed and financially non-distressed samples. The optimal number of clusters for both samples are decided by the WB index. However, the major drawbacks of this method are the high possibility of over-fitting the training data as well as the time consumed to find the optimal results. Ruibin et al. evaluated the effectiveness of ML approaches for predicting distress in publicly listed Chinese businesses (Geng et al. 2015). They examined three popular classifiers in data mining and analyzed their effectiveness using majority voting. A study on datasets gathered from various nations is also observed utilizing data mining techniques in (Bae 2012). SVM and NN classifiers show fairly good efficiency in predicting financial distress.

The learning time of NN classifiers was substantially decreased with the introduction of parallel processing technology. As a result, researchers have attempted to form DNN-based models for prediction purposes (Huang and Yen 2019; Shen et al. 2015). Due to their exceptional classification performance, DL algorithms were eventually used in the field of FDP (Glorot et al. 2011). Ribeiro and Lopes presented a DBN-based prediction model to forecast failures in French companies (Ribeiro and Lopes 2011). Matin et al. (2019) present distress prediction using DL, which employs unstructured textual information

in financial statements for prediction. To improve prediction performance on corporate financial problems, a genetic algorithm was combined with existing machine learning techniques. Jie Sun used the genetic algorithm in conjunction with the Decision Tree to optimize the financial ratio defined in the prediction of distress (Sun and Hui 2006). Hou, in 2016, introduced the K-clustering algorithm based on the genetic algorithm to solve certain conventional K-means clustering problems whilst predicting financial distress (Hou 2016). Kim et al. developed a genetic approach to optimize several heterogeneous design variables of SVMs at the same time (Kim et al. 2019). Though deep NN models are a great tool for predicting financial difficulties, there are numerous disadvantages to using them. The capacity of NN models to describe the final decision that models obtain is limited. Although NNs can give a substantial solution to the goal problem, they cannot provide a good explanation for the projected results. Shin and Lee presented a hybrid strategy of merging ANN and evolutionary algorithms and extracting rules from the bankruptcy prediction model to overcome this difficulty (Shin and Lee 2002). Another issue is that, like other NN models, an RNN contains numerous parameters that the researcher must tune. However, due to time and computational constraints, it is not possible to skim over a parameter space in order to find the optimal setting. Jang et al. (2019) developed a business failure prediction model based on LSTM and showed that it outperforms FNN and SVM in predicting construction contractors' business failure. Halim et al. (2021) examined the effectiveness of DL models such as RNN, GRU, and LSTM for FDP among publicly listed organizations in Malaysia using time-series data only in a single year. El-Bannany et al. (2020) employs three distinct DL models: MLP, LSTM, and CNN for FDP considering single-year data.

From the previous studies, it is evident that not much research has been carried out considering the problem as a time series to capture long-term dependencies from financial time-series data. We have selected an effective algorithm for time-series analysis, the LSTM-RNN algorithm. In order to make up for the shortcomings of a single prediction model, a new hybrid model that incorporates LSTM and GA is presented in this work. To the best of authors' knowledge, this is the first study that uses a hybrid optimized GA and LSTM model in FDP studies based on time-series data and predicts distress more than a year in advance. Because we only incorporated within-company data, our model is compatible with any region data. In contrast to other previous studies that predicted financial distress probability only within one year, this paper gives insight into building a time-series model and forecasting distress far in advance of its occurrence. Using GA, we focus on optimizing architectural aspects for improved performance. By adjusting the hyperparameters, we use the evolutionary algorithm approach to model the optimal network based on prediction accuracy.

## 3. Proposed GALSTM-FDP Model Formation

In this section, we provide an overview of the main models used in this study, including the LSTM, GA, and the proposed hybrid GALSTM-FDP model. The goal of this study is to identify the optimal LSTM parameter values for predicting financial distress using a genetic algorithm. We concentrate on fine-tuning the hyperparameters—the number of hidden layers, neurons, epochs, and batch size. Following this phase, the top five networks are identified and assessed based on prediction accuracy. The suggested hybrid model is then compared to traditional machine learning techniques in the following part of the research.

### 3.1. Long Short-Term Memory (LSTM)

Hochreiter and Schmidhuber proposed the LSTM neural network, which is commonly used to process sequence information due to its benefits in recognizing long-term dependencies (Hochreiter and Schmidhuber 1997). As a result, creating an LSTM model for financial time-series data is theoretically possible. An LSTM network is a type of deep RNN model composed of LSTM units. As discussed earlier, RNN is a DL network with internal feedback between neurons. The structure of RNN, as shown in Figure 1, can

theoretically map from all prior inputs to each output. The recurrent connections in RNN allows the memory of earlier inputs to survive in the internal state of the network and therefore impact the output of the network. Unfortunately, the RNN struggles from the vanishing gradients issue, that is, the impact of an input datum on a hidden layer and thus on the network output, which either deteriorates or explodes exponentially while looping through the recurrent connections of the network. Hochreiter and Schmidhuber (1997) advocated for LSTM for a lengthy span of time to address this issue. Its architecture is made up of LSTM memory blocks that are comparable to the RNN's hidden neurons. They are capable of learning long-term dynamics and avoiding the disappearing and inflating gradient issues. The LSTM-RNN can understand temporal and sequential patterns from time-series or data sequences. Many recent research have used LSTM to forecast time-series data and demonstrated that it outperforms other methodologies. To anticipate the sudden stochastic fluctuation of the financial market, a novel hybrid approach with the power of fractional order derivative was provided with the properties of LSTM networks in (Bukhari et al. 2020).

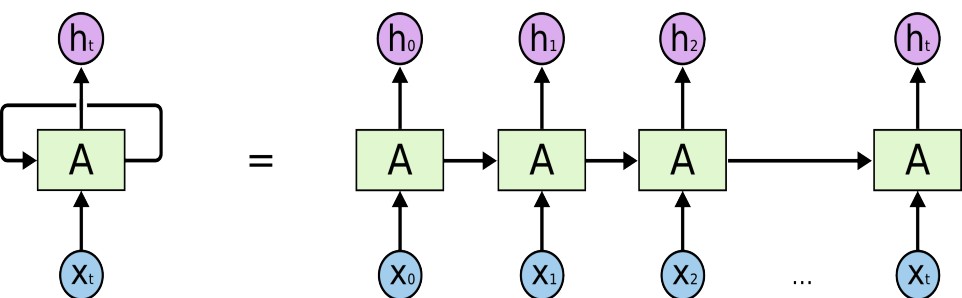

**Figure 1.** Structure of an RNN.

### 3.2. Genetic Algorithm (GA)

GA is a metaheuristic and stochastic optimization algorithm inspired by the process of natural evolution (Sun and Hui 2006). They are widely used to find near-optimal solutions to optimization problems with large search spaces. GA is a powerful optimization technique that works on principles such as selection, crossover, and mutation. Processing the GA can be divided into different stages: initialization, fitness calculation, termination condition check, selection, crossover, and mutation.

### 3.3. Hybrid GALSTM-FDP Model for FDP

This study develops an FDP model using an LSTM-RNN coupled with GA to obtain the best parameter setting for our problem. As previously stated, an LSTM is a form of RNN designed to learn sequential and temporal patterns from time-series data. The proposed GALSTM-FDP structure contains one input layer with time step = 3, one or more hidden layers, a dropout layer for each LSTM layer, and an output layer. The flow diagram of the proposed model is given in Figure 2.

A dropout layer should be added to every LSTM layer. By neglecting randomly chosen neurons during training, this layer alleviates over-fitting and therefore decreases sensitivity to the particular weights of individual neurons. The 20% rule is frequently employed as a reasonable balance between maintaining model accuracy and avoiding over-fitting.

We have used a tumbling window rather than a sliding window as our data are panel data that consider each company distinctly. For this, we have set our stride parameter to 5 and window length to 3. As we are predicting distress two years ahead, the number of input is 3 (taking the first 3 years in the dataset and skipping the next year) with 26 features.

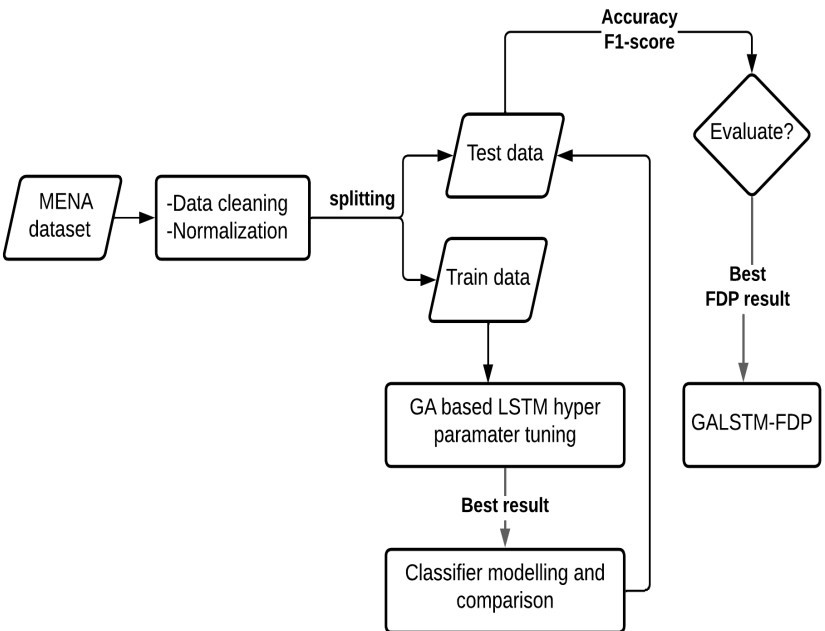

**Figure 2.** Flow diagram of the GALSTM-FDP model for FDP.

Optimizing the LSTM Model Using GA

This study examines how GAs can be used to optimize LSTM networks for FDP. We have applied GA to optimize the number of units in the LSTM layers, activation functions, number of epochs, and batch size. The algorithm starts with a random initial population, forming the core of the GA. At each generation, a new population is created from the initial population after the process of evaluation, selection, crossover, and mutation. This process continues over the generations until an optimum solution is attained. Figure 3 depicts the process of optimizing the LSTM model using GA.

- Generate the initial population: The creation of the initial population, which forms the basis of GA is the first step. By randomly mixing various hyperparameter values, 20 distinct LSTM networks are produced, constituting the starting population of the GA. Antonio Dourado (2013) has offered a solid rule of thumb for avoiding over-fitting: start with [Number of Training Samples/2 × (Number of Input Neurons + Number of Output Neurons). In most cases, it works, but if the problem is too simple or complex, we can experiment increasing or decreasing the number of neurons. We have picked two values that are less than 64 and two values that are larger than 64. The popular Adam optimizer was chosen as the activation function for this model. The initial population in this study is made up of a collection of 20 random networks. The other two parameters that can be varied are the number of epochs and batch size.

- Compute fitness: Each individual network in the population is trained and tested using the MENA dataset and scored according to the predictive accuracy. We did not run cross-validation on this dataset because it is a time-specific dataset. The highest-scoring networks are retained in order to increase the population of the next generation. The remaining networks of the existing population are discarded.

- Selection: An LSTM network with high accuracy has a greater probability of being chosen for the following generation. During the selection phase, we identify networks from the present population that will be passed down unchanged to the future generation. The top-ranked quarter (five numbers) networks based on accuracy scores are transferred directly to the next population. In order to prevent being trapped at the local maximum, three low-performing networks are also preserved and transferred to the next generation.

- Breeding: For the next generation, we currently have eight networks in the population. The remaining 12 are the product of crossover or breeding. To produce one or more children for the following generation, two networks from the current population, known as parents, are required. The parents are chosen based on their scores, and the network parameters are combined to generate a new offspring that is a hybrid of its parents. Each child in this study is a network with a random set of parameters from its parents.
- Mutation: We randomly adjust some of the properties of random networks in the population to have a population for the next generation. This technique aims to create better networks for the population.

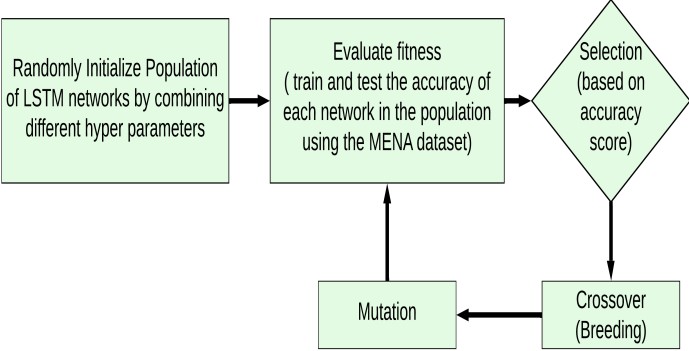

**Figure 3.** Optimizing the LSTM model using GA for FDP.

Algorithm 1 involved in this study is given below:

---

**Algorithm 1** GALSTM-FDP

---

1: *Construct a population of LSTM networks and allocate them all random hyperparameters.*
    *- Number of hidden layers: {1, 2, 3, 4}*
    *- Neurons per layer: {16, 32, 64, 128, 256}*
    *- Batch size: {35, 70, 105, 140, 175 }*
    *- Epochs: {50, 100, 200, 300, 400, 500}*
    *- Select 20 random LSTM as the current population.*
2: *Train each network in the population using the given MENA dataset.*
3: *Calculate the prediction accuracy for each one and pick the top performing ones for the next step.*
4: *Select some top networks and non-top networks (to avoid getting stuck at local maximum).*
5: *Crossover: The parameters of two members of the chosen networks are crossed over. By merging the parents, this will result in a "child" network possessing some features of the first and some of the second.*
6: *Mutate the parameters of some of the child networks*
7: *Save the generated child networks in a new population and allocate it to the variable that holds the former population.*
8: *Repeat steps 2 to 7 for 10 generations.*

---

We used an initial set of 20 random networks to establish the population in this study, and we repeated the evaluation, selection, crossover, and mutation procedure for 10 generations. As a result of our research, we have trained approximately 200 LSTMs, resulting in a stronger population over time. In the following part, the top five LSTM networks from the final population set are determined and examined based on prediction accuracy. The predictive accuracy of networks in the population improved over generations, and the models were trained and generated using the Python libraries Scikit-learn and Keras. We also evaluated the performance of the proposed model to that of traditional ML models. The simulation results revealed that the proposed hybrid model has much greater prediction accuracy when compared to SVM, DT, and standalone LSTM classifier algorithms.

## 4. Data and Modeling

The goal of this study is to identify the optimal LSTM parameter values for predicting financial distress using a genetic algorithm. We have applied GA to optimize the number of units in the LSTM layers, activation functions, number of epochs, and batch size. Following this phase, the top five networks are identified and assessed based on prediction accuracy. The suggested hybrid model is then compared to traditional machine learning techniques in the following part of the research. To evaluate whether a company shows up as financially distressed or not, the FDP problem discussed in this paper is modeled as a binary classification problem. In the financial dataset, the output or target attribute belongs to any one of the two classes: financially distressed companies and financially healthy companies. All of the other attributes in the dataset are continuous values with the exception of this target attribute, which has a binary value.

*Dataset Description:* The dataset considered for evaluating the predictive performance includes sample data gathered from MENA (Middle East and North Africa) listed firms, extracted from the Osiris database. Financial companies are exempted due to the difference in their operational environment. We selected only the companies with 5 years (2015–2019) of consecutive data in the final dataset, totaling to 9765 company years of 1953 companies. The financial variables extracted from financial statements and balance sheets of respective companies are selected considering the variables and ratios used in prior studies in this area. A detailed description of the ratios calculated from 20 financial variables is given in Table 1. The 20 variables along with 6 ratios are taken as input to the network.

**Table 1.** Financial Indicators.

| Indicators | Formula |
| --- | --- |
| *Solvency* | Total liabilities/Total assets |
| | Current assets/Current liabilities |
| | Current assets-inventory/Current liabilities |
| | Total liabilities/total shareholders' equity |
| | Current liabilities/total assets |
| | Net operating cash flow/current liabilities |
| | Earnings before interest and tax/interest expense |
| *Capital Expansion* | Net profit/number of ordinary shares at the end of year |
| | Net assets/number of ordinary shares at the end of year |
| | Net increase in cash and cash equivalents/number of ordinary shares at the end of year |
| | Capital reserves/number of ordinary shares at the end of year |
| *Profitability* | (Sales revenue–sales cost)/sales revenue |
| | Net profit/sales revenue |
| | Earnings before income tax/average total assets |
| | Net profit/average total assets |
| | Net profit/average current assets |
| | Net profit/average fixed assets |
| | Net profit/average shareholders' equity |
| *Business Development* | Business income of this year/Business income of last year |
| | Total assets of this year/total assets of last year |
| | Net profit of this year/net profit of last year |
| *Operational Capabilities* | Main business income/average total assets |
| | Sales revenue/average current assets |
| | Sales revenue/average fixed assets |
| | Main business cost/average inventory |
| | Main business income/average balance of accounts receivable |
| | Cost of sales/average payable accounts |

**Table 1.** *Cont.*

| Indicators | Formula |
|---|---|
| *Structural Soundness* | Current assets total assets |
| | Fixed assets/total assets |
| | Shareholders' equity/fixed assets |
| | Current liabilities/total liabilities |

Prior to actual fitting of the model to the dataset, three steps must be completed: (1) Data cleaning, (2) Normalization, and (3) Splitting.

1. Data cleaning: This phase is used to validate the data contained within the dataset. As seen in Table 2, the value ranges between variables have a wide variation. Our samples, like most real datasets, contain null values and missing properties. These issues were resolved during the pre-processing stage, where we scaled the dataset after filling in the missing values with forward and backward fill within each company as the units of several variables were also different.
2. Normalization: The model rescales the variables to a range of −1 to 1 to produce a fair result. This process is needed to convert all of the column values in the dataset to a common scale without distorting the value ranges or causing data loss.
3. Dataset samples splitting: The dataset was then turned into the time-series data that the LSTM network required. For a realistic evaluation of performance, datasets must be divided into testing and training sets. We cannot partition our data like other non-time specific datasets because they are time series. At any given time, our network receives 5 years' worth of data (first 3 years of data from 5 years as input and fifth year as output). Considering this, we have divided our training set into a multiple of 5: 7810 samples in the training set and the remaining 1955 years in the testing set. This is approximately 80% of data in the training samples and the remaining 20% in the testing samples.

**Table 2.** Descriptive statistics of all individual financial attributes.

| Variables | Mean | Min | Max |
|---|---|---|---|
| Total Liabilities | 88,751.36508 | 13 | 1,669,220 |
| Total Assets | 183,280.635 | 5763 | 3,091,702 |
| Current Assets | 43,249.87 | 138 | 676,520 |
| Current Liabilities | 46,492.642 | 10 | 790,074 |
| Accounts Receivable | 11,041.079 | 5.63 | 200,071 |
| Accounts Payable | 11,966.0634 | 205 | 374,494 |
| Total Shareholders Equity | 93,131.111 | 5565 | 1,422,482 |
| Net Cash Flow | 17,297.667 | −4600 | 481,539 |
| EBIT | 8274 | 16 | 267,461 |
| Net Profit | 4253.9524 | −31,571 | 156,702 |
| Cash and Cash Equivalent | 12,907.0158 | 19 | 209,716 |
| Cost of Goods Sold | 22,169.1746 | 345 | 266,764 |
| Sales | 1,730,068.53 | 93 | 10,301,478 |
| Shares | 402,708 | 30,318.8 | 3,901,347 |
| Capital Reserves | 5774.65 | −615 | 182,827 |
| Fixed Assets | 140,030.76 | 599 | 2,415,182 |
| Average Total Assets | 274,315.7778 | 8815 | 4,608,537.5 |
| Average Current Assets | 66,022.484 | 210 | 1,048,037 |
| Average Fixed Assets | 208,293.2937 | 825 | 3,560,500 |
| Average Equity | 140,519.53 | 8476 | 2,227,284 |
| Average Accounts Receivable | 13,773.69 | 4069 | 327,807 |
| Average Accounts Payable | 16,861.05 | 794 | 555,164.5 |
| Net Increase in Cash | −3622.984 | −151,247 | 16,129 |

## 5. Empirical Analysis

Our model is built on the popular Keras ML library in Python. We have used a tumbling window rather than a sliding window as our data are panel data that consider each company distinctly. For this, we have set our stride parameter to 5 and window length to 3. As we are predicting distress two years ahead, the number of input is 3 (taking the first 3 years in the dataset and skipping the next year) with 26 features.

To evaluate whether a company shows up as financially distressed or not, the FDP problem discussed in this paper is modeled as a binary classification problem. In the financial dataset, the output or target attribute belongs to any one of the two classes: financially distressed companies and financially healthy companies. All of the other attributes in the dataset are continuous values with the exception of this target attribute, which has a binary value.

In this research, *negative* are companies that are financially distressed, whereas *positive* are those that are financially sound. If the F1-score is employed as a performance metric, a balance between precision and recall could be attained.

### 5.1. FDP Performance Evaluation Metrics

We compare the classifiers' performances in predicting financial distress with two common ML evaluation metrics:

1.  Accuracy: For performance assessment, the accuracy of the test data is considered. Accuracy is the percentage of the number of correct predictions from all the predictions made.
2.  F1-score: F1-score is a function of precision (Equation (1)) and recall (Equation (2)) as defined by Equation (3).

$$Precision = \frac{True - Positive}{(True - Positive + False - Positive)} = \frac{True - Positive}{(Total - Predicted - Positive)} \tag{1}$$

$$Recall = \frac{True - Positive}{(True - Positive + False - Negative)} = \frac{True - Positive}{(Total - Actual - Positive)} \tag{2}$$

$$F1 - score = 2 * \frac{(Precision * Recall)}{(Precision + Recall)} \tag{3}$$

### 5.2. Correlation and Feature Importance

The descriptive statistics of the attributes are given in Table 2. Figure 4 shows the Pearson's correlation, and Figure 5 depicts the feature importance of the attributes, respectively. A high correlation means values between $-0.50$ and $-1.00$. Feature importance order is depicted in Figure 5. We included all the attributes, as omitting the least important ones resulted in lower performance; extreme outliers in the dataset were detected, analyzed (e.g., to determine whether they resulted from errors in data entry), and removed.

### 5.3. Analysis of the Proposed Hybrid GALSTM-FDP Model for FDP

This section presents and analyzes the findings of the proposed hybrid GALSTM-FDP model for FDP. The average accuracy values of all 20 models in the population over 10 generations are shown in Figure 6.

According to the graph, the financial distress prediction accuracy of networks in the population has increased over generations. This means that the GA optimizes the network at each iteration, resulting in the best solution for FDP at the completion of the iterations. At each generation, a set of 20 networks is trained and scored, then crossed over and mutated to produce the new population of networks for the coming generation. Table 3 displays the parameters of the five highest scoring networks based on the percentage of prediction accuracy and F1-score. Figure 7 depicts the accuracy and F1-score graphs for the same.

The highest predictive accuracy and F1-score for the dataset in percentages are 91.50 and 92.67, respectively. A dataset with a greater number of attributes can be trained more effectively and provide a more robust model than a dataset with fewer attributes. From the

above table, we can see that although we have included three and four hidden layers for analysis the best performing was two layers. That is, an LSTM with two hidden layers is enough for our FDP problem. Training the model for more number of epochs did yield a higher accuracy. Our input layer had 26 input attributes, which in turn is the shape of the input layer. Moreover, 64 or 128 neurons per layer gave a better performance than the others. We set the batch sizes as multiples of 5 because we take 5 years of data as a single input; 35 or 70 inputs per batch performed well in our setting.

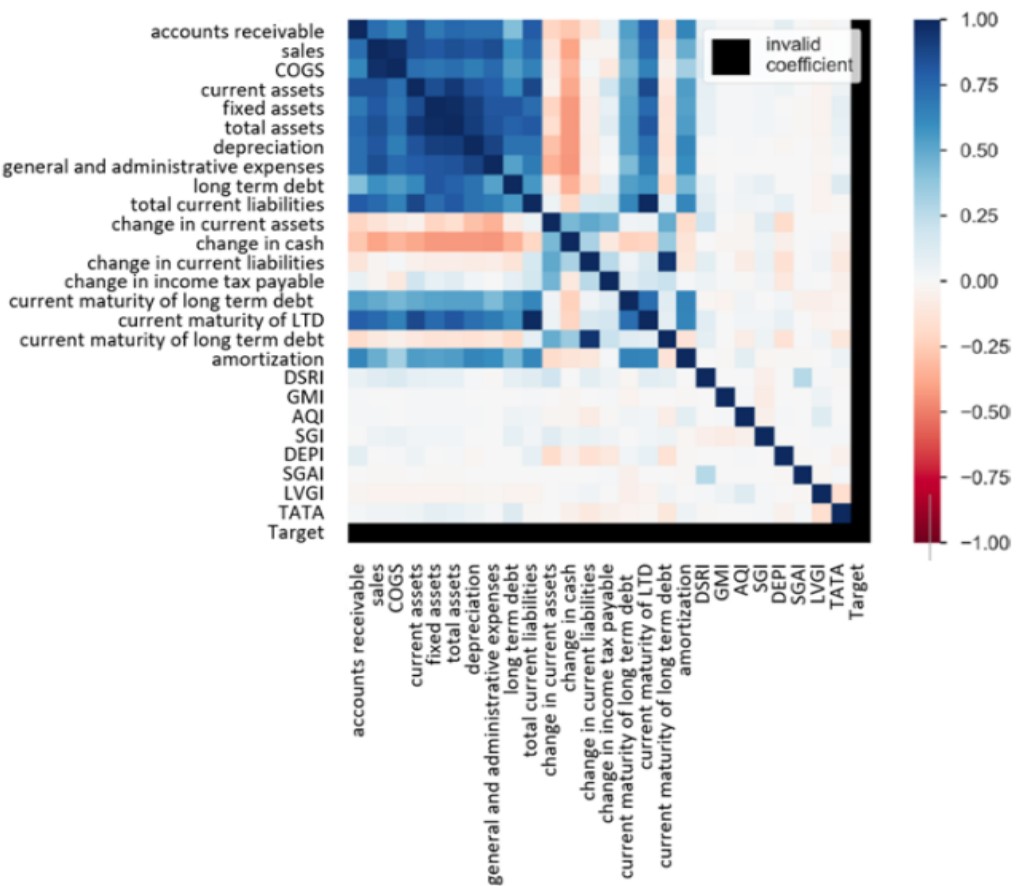

**Figure 4.** Pearson's r correlation.

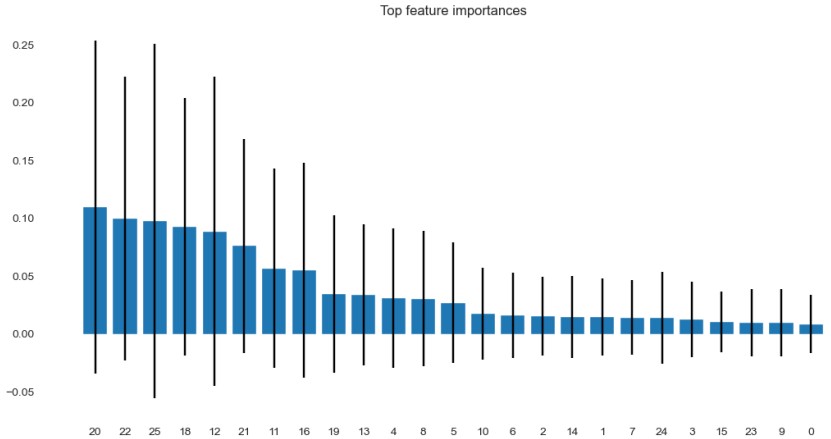

**Figure 5.** Feature importance of the attributes.

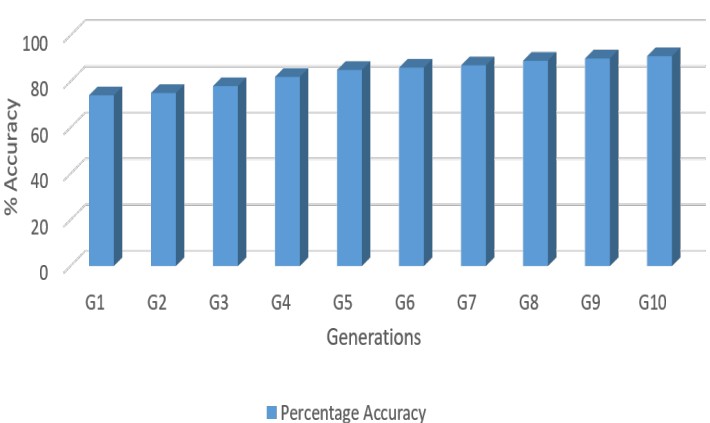

**Figure 6.** Average accuracy score of the GALSTM-FDP model across 10 generations.

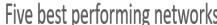

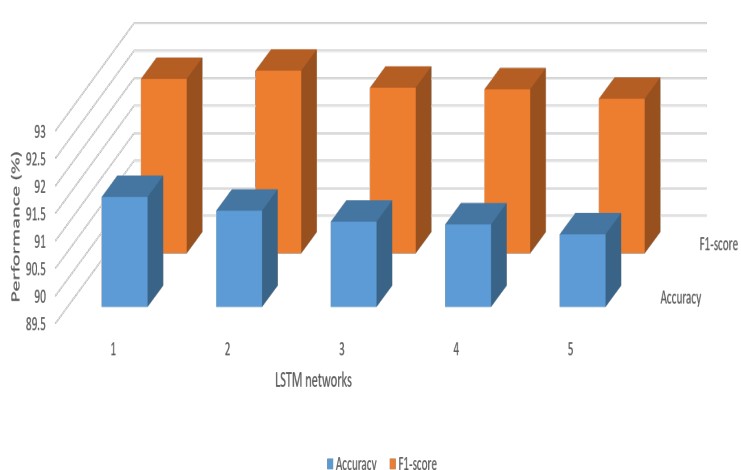

**Figure 7.** Best-performing 5 LSTM models.

**Table 3.** GALSTM-FDP: Network parameters of top 5 networks based on percentage of accuracy and F1-score.

| Network | Hidden Layers | Neurons per Layer | Batch Size | Epoch | Testing Accuracy (%) | Testing F1-Score (%) |
|---------|---------------|-------------------|------------|-------|----------------------|----------------------|
| 1 | 2 | 64 | 35 | 500 | 91.50 | 92.67 |
| 2 | 2 | 64 | 70 | 400 | 91.25 | 92.82 |
| 3 | 1 | 64 | 35 | 500 | 91.05 | 92.51 |
| 4 | 2 | 128 | 35 | 400 | 91.00 | 92.48 |
| 5 | 1 | 128 | 70 | 500 | 90.82 | 92.31 |

In the last stage of the research, we compared our improved LSTM network against SVM, DT, and basic LSTM classifier algorithms. Prediction outcomes in terms of accuracy and F1-score are shown in Table 4. The statistical findings show that the suggested optimized GALSTM-FDP model's prediction accuracy is substantially greater than that of the traditional ML models.

**Table 4.** Comparison of GALSTM-FDP, LSTM, SVM, and DT in solving the FDP problem.

| Classifiers | Financial Distress Prediction Performance | |
| | Accuracy | F1-Score |
| --- | --- | --- |
| *GALSTM-FDP* | 91.50 | 92.67 |
| *LSTM* | 88.75 | 90.46 |
| *SVM* | 86.50 | 89.67 |
| *DT* | 81.75 | 85.46 |

*5.4. GALSTM-FDP: Overall Analysis*

The GALSTM-FDP model's greater performance can be attributed to the fact that the architecture of the LSTM network substantially improved learning efficiency and reduced the need for unnecessary computations. We integrated GA with LSTM to find the optimum hyperparameter configuration for LSTM with the aim of optimizing architectural aspects of the model for enhanced prediction power. The application of GA to investigate the optimal architectural factors derives results through this genetic search. The findings imply that proper parameter tuning is a necessary precondition for achieving adequate performance. Even if deep learning algorithms are expanding quickly, finding the ideal set of deep architecture parameters requires extensive understanding. The experimental findings, however, indicate the potential for the hybrid GALSTM-FDP model's application in FDP and show that it can be a useful tool for finding the best or nearly best results. Because we incorporated just within-company data, our model is compatible with any region data. In other words, it is most effective in handling FDP based on time-series data, can accurately predicts distress two years ahead, and is beneficial for the corporate, market participants and policymakers.

**6. Conclusions and Future Work**

Finding the best-performing FDP model has always been a focus of researchers, and numerous FDP models have been developed since then. We presented a hybrid GALSTM-FDP model for FDP and assessed the performance of LSTM by adjusting the hyperparameters using a GA. It was observed that if the model is configured with one or two hidden layers, an acceptable predicted performance rate can be obtained. We trained and evaluated the models with a dataset of 26 input variables collected from 1953 firms in the MENA area. The simulation results show that the suggested model outperforms standalone LSTM and other conventional ML methods such as SVM and DT. All of the variables in our analysis are within-firm variables. To improve the model, macroeconomic and other industrial aspects may be incorporated in future study.

**Author Contributions:** All authors contributed together to realize this work, and their exact contributions are difficult to specify. Conceptualization and methodology by A.A.A., S.K., and A.M.K.; Software and original draft preparation by A.A.A. and M.E.B.; Validation, writing, reviewing, editing, and supervision by M.E.B., S.K., and A.M.K. All authors have read and agreed to the published version of the manuscript.

**Funding:** No funding was obtained for this study.

**Informed Consent Statement:** Not applicable.

**Data Availability Statement:** Available on request.

**Conflicts of Interest:** The authors declare that they have no competing interests.

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
