# Peer review of "GALSTM-FDP: A Time-Series Modeling Approach Using Hybrid GA and LSTM for Financial Distress Prediction"

_ijfs, doi:10.3390/ijfs11010038_

Round 1
Reviewer 1 Report
The authors have written an article on Financial Distress Prediction based on LSTM + GA. Please address the following points.
1. References/citations are missing. Check line no. 23 and 217.
2. Figure 3. shows financial indicators. Please mention dependent and independent variables in the dataset. What is the correlation among variables?
3. Does the Problem of financial distress prediction belong to a classification or regression category?
4. There seems to be confusion about the outcome of the prediction. As per the results, you have used accuracy and F1 score as evaluation metrics suitable for evaluating a classification model. But what are you trying to predict? An Example or sample use case or scenario is to be included in the article.
5. Line no. 287 mentions that missing values are taken care. Which methods are considered for handling missing values?
6. Include a section on exploratory data analysis of the dataset to add more clarity and increase the readability of the article.
7. What is the significance of Figure 4. It is obvious right, as the number of iterations increased, accuracy seems to increase. Why did you include this graph?
8. Please add a section in results section to emphasize the importance of the hybrid algorithm. Especially what happens if you don’t use GA for hyperparameter tuning. Some of the well known methods for hyperparameter tuning are manual search, grid search, randomized search, halving grid search, halving randomized search, hyperopt-sklearn, bayes search, etc. Why did authors not use these methods and instead chose GA? Justify.
9. Introduction explains financial distress, but later sections focus more on LSTM without emphasizing how LSTM+GA assist in FDP.
Overall there are ample of scope for major improvements.
Author Response
Dear Professor
The authors would like to thank you for your valuable comments and suggestions. We have carefully implemented the recommended modifications. We believe that the new version addresses the reviewer's concerns
Please find attached the response report.

Reviewer 2 Report
The manuscript can be published after some minors. Please consider:
Please compare the existing literature with the proposed algorithm. Also consider https://doi.org/10.1080/02664763.2020.1823947 which uses SVM to predict bubbles.
The reference in line 23 is missing.
Please provide theoretical background of the used algorithms.
Please compare the performance of the model with standalone LSTM and other machine learning algorithms.
Please correct the style between the lines 248 and 258.
The resolution of Figure 3 must be increased indeed it is not a figure it is a table.
Author Response

(The authors gave the same response as above.)

Round 2
Reviewer 1 Report
The authors have addressed the comments.